

# Signal contribution of distant areas to cosmic-ray neutron sensors – implications on footprint and sensitivity

Martin Schrön[1], Markus Köhli[2,3], and Steffen Zacharias[1]

[1]UFZ - Helmholtz Centre for Environmental Research GmbH, Leipzig, Germany
[2]Physikalisches Institut, Heidelberg University, Heidelberg, Germany
[3]Physikalisches Institut, University of Bonn, Bonn, Germany

**Correspondence:** Martin Schrön, martin.schroen@ufz.de

**Abstract.**

This paper presents a new theoretical concept to estimate the contribution of distant areas to the measurement signal of cosmic-ray neutron detectors for snow and soil moisture monitoring. The algorithm is based on the local neutron production and their transport mechanism, given by the neutron-moisture relationship and the radial intensity function, respectively. The purely analytical approach has been validated with physics-based neutron transport simulations for heterogeneous soil moisture patterns, exemplary landscape features, and remote fields at a distance. We found that the method provides good approximations of simulated signal contributions in patchy soils with typical deviations of less than 1 %. Moreover, implications of this concept have been investigated for the neutron-moisture relationship, where the signal contribution of an area has the potential to explain deviating shapes of this curve that are often reported in literature. Finally, the concept has been used to develop a new practical footprint definition to express whether or not a distant area's soil moisture change is actually detectable in terms of measurement precision. The presented concepts answer long lasting questions about the influence of distant landscape structures in the integral footprint of the sensor without the need for computationally expensive simulations. The new insights are highly relevant to support signal interpretation, data harmonization, and sensor calibration, and will be particularly useful for sensors positioned in complex terrain or on agriculturally managed sites.

**Keypoints**

- The signal contribution of distinct patches in the footprint can be estimated analytically without dedicated simulations,

- Learning about the contribution of individual landscape features may explain the shape of the neutron-moisture relationship,

- A new practical footprint definition expresses whether or not remote soil moisture changes are visible in the CRNS signal.



## 1 Introduction

Cosmic-Ray Neutron Sensing (CRNS) is an established measurement technique for water content in soils and snow (Andreasen et al., 2017). The high integration depth and the large measurement footprint have been shown to provide an important advantage for field-scale applications compared to conventional point-scale sensors. However, the intrinsic integration over the whole footprint volume conceals the individual contributions of different patches and may result in biased observations (Franz et al., 2013; Schrön et al., 2018; Schattan et al., 2019).

The footprint has been initially characterized by its radius of around 300 m by Zreda et al. (2008) and Desilets and Zreda (2013) without significant dependency on soil moisture or air humidity. Later, Köhli et al. (2015) revisited the physical assumptions of the underlying neutron simulations and proposed a moisture-dependent footprint radius in the range from 130 m to 240 m. Besides the epithermal neutron transport, also thermal neutron footprints were investigated by Jakobi et al. (2021). These studies take into account the high complexity of neutron transport physics, which usually can only be investigated with computationally expensive Monte-Carlo simulations. The accepted definition of the epithermal footprint radius, $R_{86}$, covers the $1 - e^{-2} \approx 86\%$ quantile of detected neutrons. This measure was introduced by Desilets and Zreda (2013) and has been inherited by Köhli et al. (2015) in order to maintain consistency. However, the definition involves four problems: (i) The radial intensity function, $W_r(h, \theta)$, does not follow a simple exponential shape (Köhli et al., 2015; Schrön et al., 2017). Therefore, the $1 - e^{-2}$ limit may be misleading when it is used to draw conclusions about the intensity-radius relationship elsewhere. (ii) High-quantile values for strongly non-linear functions may overestimate the long-range influence of neutrons, regardless of how often and where they have probed the soil. (iii) The fact that the definition of the footprint has been developed for homogeneous situations increases the uncertainty and the interpretation of more heterogeneous and more complex terrain. And (iv), the definition hardly allows to investigate problems and questions that often arise during practical applications: Is the detector sensitive to remote soil moisture changes? Does a certain patch of the area influence the detector signal? By how much?

As a standard solution for such questions, neutron transport physics-based Monte Carlo codes could be employed with detailed modeling of the local conditions (as has been done by, e.g., Franz et al., 2013; Köhli et al., 2015; Schrön et al., 2018; Schattan et al., 2017; Li et al., 2019, among others). However, this technique is impractical for quick assessments and mostly limited to scientific applications.

While cosmic-ray neutron sensors are usually employed to track soil moisture changes in the area of their footprint, complex structures or heterogeneous patterns in the footprint may influence the measurement unexpectedly. The dependency of the measured neutrons on soil moisture changes has been originally expressed by the neutron-moisture relationship (Desilets et al., 2010; Köhli et al., 2021) and has also been adapted for snow (Schattan et al., 2017). Many natural sites are highly heterogeneous and thus knowledge of the contribution of distant areas to the measurement signal would be very useful, e.g. to support calibration sampling, sensor location design, data interpretation, and uncertainty assessment. Typical events modulating water abundance and distribution are, for example, land management activities like harvesting (Franz et al., 2016; Song et al., 2019), plowing (Kasner et al., 2022), or irrigation (Li et al., 2019; Ragab et al., 2017), or natural events like rain water interception in



forests (Baroni and Oswald, 2015; Andreasen et al., 2017; Schrön et al., 2017) , snow melt and redistribution (Schattan et al.,
2019), or different soil dry-out rates due to different hydraulic conductivity (Scheiffele et al., 2020).

In the past, spatially (and temporarily) variable factors within the footprint influencing the neutron signal have often been
identified as the source of unexplained features in the data. These discoveries sometimes boosted scientific insights on neutron
transport, and even led to more reliable hydrological data (see, e.g., Bogena et al., 2013; Schrön et al., 2017, 2018; Schattan
et al., 2019; Rasche et al., 2021). However, at most heterogeneous sites, CRNS calibration and validation remains a challenge,
since the influence of the differing structures or patches in the footprint to the signal is usually not known (Coopersmith
et al., 2014; Lv et al., 2014; Iwema et al., 2015a; Franz et al., 2016; Heistermann et al., 2021). For this reason, many authors
reported differing shapes of the neutron-moisture curve and conducted site-specific empirical re-parameterizations to fit their
data (Rivera Villarreyes et al., 2011; Lv et al., 2014; Iwema et al., 2015b; Heidbüchel et al., 2016). Others developed directional
sensors to focus the measurement only on specific parts of the landscape (Francke et al., 2022), which remains a technological
challenge on its own.

One way to approach the estimation of signal contributions of different areas in the footprint is to use the radial intensity
function $W_r$. First attempts to realize this idea have been performed by Schrön et al. (2017), who improved the sensor calibra-
tion by applying different weights to areas depending on size, distance, and landuse class, and also by Schrön et al. (2018), who
excluded the contribution of a concrete area around a grass land site in order to improve reliability of soil moisture dynamics
measured by stationary CRNS.

In the present study we aim at generalizing this concept for typical combinations of heterogeneous land use and soil moisture
patterns. Our hypothesis is that the contribution to the detector signal of various complex areas in the footprint can be estimated
analytically based on the existing theories about neutron production and transport. The first section will describe the proposed
concept and discuss its potential limitations. Then, the concept will be evaluated by dedicated neutron transport simulations
for various scenarios of different soil moisture patterns, land-use types, and geometries. We further aim at exploring two
applications of this concept, first, to assess its explanatory power for the shape of the neutron-soil moisture relationship. And
second, to provide a more practical footprint definition expressing whether or not a distant area's soil moisture change (e.g.,
by irrigation or rainwater interception, or faster drainage) is actually visible to the neutron signal in terms of measurement
precision.



## 2 Methodological concept

### 2.1 The radial intensity function

The sensitivity of a central detector to an infinitesimal ring at distance $r$ has been described by Köhli et al. (2015) and refined by Schrön et al. (2017) as:

$$W_r(h,\theta,P,H_{\mathrm{veg}}) \sim F_1\,e^{F_2 r} + F_3\,e^{F_4 r}, \tag{1}$$

which is a combination of two exponential functions with factors and slopes ($F_{1\ldots4}(h,\theta,P,H_{\mathrm{veg}})$) that represent the complex nature of neutron transport in humid environments. This *radial intensity function* $W_r$ (see Fig. 1a) depicts the number of detected neutrons that originated in the soil at the distance $r$ under certain homogeneous conditions of air humidity $h$, (soil) water equivalent $\theta$, air pressure $P$, and vegetation height $H_{\mathrm{veg}}$. It can also be expressed as $W_{r^*}(h,\theta)$ with $r^* = r(P, H_{\mathrm{veg}})$ being scaled by air pressure and vegetation influence (see Schrön et al., 2017, for the details). We use this simplified formulation in the following, while results can be easily transferred to other pressure and vegetation conditions by rescaling $r$ as mentioned.

An alternative parameterization, $W_r^* \approx W_r(h,\theta)$, has been proposed by Schrön et al. (2017) as an approximation for average humidity and soil moisture conditions:

$$W_r^* = \left(30\,e^{-r/1.6} + e^{-r/100}\right) \cdot \left(1 - e^{-3.7 r}\right). \tag{2}$$

This approximation can be evaluated in a computationally more efficient way, as it does not depend on humidity and soil moisture, but at the same time it is less accurate towards the extreme ends of dry or wet conditions.

The integral of $W_r(h,\theta)$ over all radii $r$ represents the total number of detected neutrons, $N$:

$$N(h,\theta) = \int_0^\infty W_r(h,\theta)\,\mathrm{d}r. \tag{3}$$

In other words, the detectable neutron intensity at the center of the radial footprint is the sum of all the ring intensities, $W_r$, across in the whole domain $\Omega$.

Based on this definition, Köhli et al. (2015) derived the hitherto accepted CRNS footprint radius, $R_{86}$, as the distance within which 86 % of all the detected neutrons originated:

$$0.86\,N(h,\theta) = \int_0^{R_{86}} W_r(h,\theta)\,\mathrm{d}r. \tag{4}$$



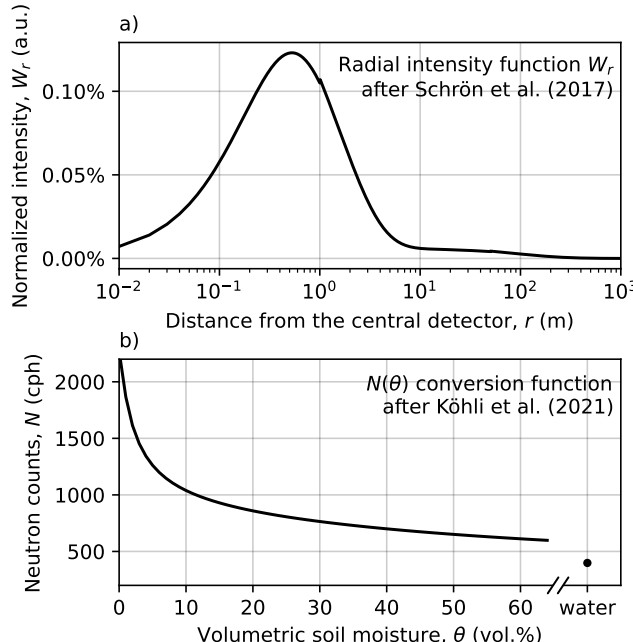

**Figure 1.** Basic functions of CRNS theory. a) The radial intensity function, $W_r(\theta = 0.10, h = 5)$, representing the intensity contribution of all points at distance $r$ to the detector signal. b) The conversion function, $N(\theta, h = 5)$, for a typical stationary CRNS sensor.

## 2.2 The concept of signal contributions from sub-domain areas

Let $A_i \in \Omega$ be a set of sub-domain areas with water content $\theta_i$, constituting the whole domain $\Omega$ (see Fig. 2 for an exemplary illustration). We propose that the total measured neutron intensity at the center, or *effective neutron intensity* $\hat{N}$, is the sum of all the neutrons which were generated in $A_i$, weighted by their ability to reach the sensor, i.e.,

$$\text{Signal from area } A_i = \underbrace{N(\theta_i)}_{\text{production}} \times \underbrace{\int_{A_i} W_r(h, \hat{\theta})}_{\text{transport}} \,. \tag{5}$$

This quantity can be expressed as the product of the locally generated neutrons, $N_i = N(\theta_i)$, and the radial intensity weight of its area, $w_i$. The total signal then is:

$$\hat{N} = \frac{\sum_i w_i N_i}{\sum_i w_i}, \quad \text{where} \quad w_i = \int_{A_i} W_r \,, \quad A_i \in \Omega \,. \tag{6}$$

In a homogeneous domain, where $N_i = N_j \, \forall \, i, j$, the integral weight $w_i$ of an arbitrary subset area $A_i$ directly represents the area's contribution to the measured neutron signal, only depending on size and distance. In an inhomogeneous scenario, the contribution depends also on the local count rate $N_i$. For example, the effective signal of a symmetrical domain containing two identical half-spaces with a sensor in the center would be an equal combination of the individual intensities, $\hat{N} = 0.5 \, N_1 + 0.5 \, N_2$.





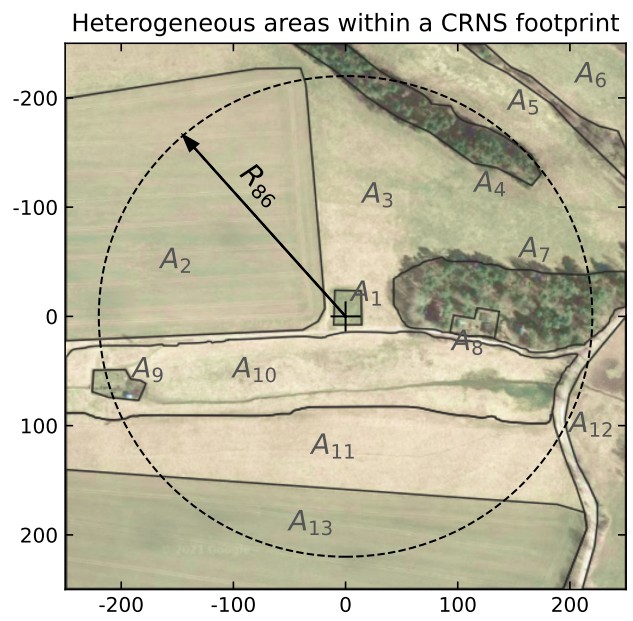

**Figure 2.** Exemplary scenario of the site "Schäfertal" (51.6551°N, 11.0525°E, Wollschläger et al., 2017). Relevant areas $A_i$ within the footprint of the central CRNS detector (+) are indicated by black borders, e.g., the agricultural land ($A_2$, $A_{13}$), forest sites ($A_4$, $A_7$), buildings ($A_8$, $A_9$), or the river creek ($A_{10}$). The dashed circle illustrates the conventional footprint definition, $R_{86}$, within which 86 % of detected neutrons probed the soil. (Satellite image by © Google Maps)

The *relative contribution* $c_i$ of the area $A_i$ to the sensor signal is particularly useful for inhomogeneous, i.e., patchy scenarios and can be expressed as:

$$c_i = w_i\, N_i/\hat{N}\,. \tag{7}$$

In the above example, the relative contribution of the field 1 would be $c_1 = 0.5\, N_1/\hat{N} = N_1/(N_1 + N_2)$.

The proposed method can be applied to an arbitrarily complex combination of areas $A_i$. The weight of the areas could be determined in two ways. In an angular environment, the integral over $A_i$ could span over the respective range of angles and radii (see Schrön et al., 2018, section 3.5):

$$\int_{A_i} W_r = \frac{1}{2\pi} \int_{\vartheta_1}^{\vartheta_2} \int_{r_1}^{r_2} W_r\, \mathrm{d}r'\mathrm{d}\vartheta'\,, \quad \text{where } A_i(\vartheta_{1,2}, r_{1,2})\,. \tag{8}$$

In gridded environments, the spatial integral is simply the sum of the weights over all grid cells (see Schrön et al., 2017, section 2.3):

$$\int_{A_i} W_r = \sum_j W_{r_j}/r_j \quad \forall j \in A_i\,. \tag{9}$$



For computational implementations, it is often easier to perform calculations on Cartesian coordinate systems (the latter option), such that the individual weight of each grid cell follows $W_r/r$.

## 2.3   Potential limitations and remarks

The simplified concept cannot claim to be able to simulate the complex physics of neutron transport for every possible scenario. Just from the way it is formulated, some potential limitations of this approach can be already expected. The performance tests in the present study aim at challenging some of these objections with synthetic examples and evaluations by Monte Carlo simulations.

A direct limitation is that $W_r$ has been initially defined as a radially symmetric function, assuming equal contribution
at distances $r$ in all directions. However, most heterogeneous regions are not radially structured, such that highly variable soil moisture patches would lead to asymmetric weights in different directions, giving the corresponding footprint radius an amoeba-like shape (see, e.g., Köhli et al., 2015; Schattan et al., 2019, Figs. 9 and 9, respectively).

The $W_r$ function has also been derived for homogeneous conditions, and thus sharp borders of soil moisture patterns may not be resolved adequately. This is particularly true in regions of high sensitivity, such as the first few meters below and around
the detector. These cases could lead to road-effect-like biases (Schrön et al., 2018) and should be avoided – while they rarely occur – in realistic applications.

Moreover, the actually detected neutron flux at the center not only depends on the neutron response of all the individual fields, but also on secondary interactions with water and soil between the detector and the remote field. These intermediate fields may influence the neutron's travel path and moderation probability since neutrons typically undergo several interactions
with the soil on their way to the detector (Köhli et al., 2015).

The concept also treats each area individually and cannot reproduce spatial interaction effects. They can occur when neutrons generated in an area typically diffuse to nearby areas, influencing their apparent local neutron intensity (see, e.g., Schrön et al., 2018). For this reason, scenarios in this study will exhibit sufficient space between distinct areas such that their neutron contribution can be assessed individually.

When it comes to evaluation of the presented concept with measurement data, it should be noted that results obtained from $W_r$-based approaches as well as from URANOS particle origins only represent detected neutrons with preceding soil contact, while a considerable fraction of the CRNS-measured signal is direct radiation from incoming neutrons (see Schrön et al., 2016, Fig. 3). This additonal signal component is usually rather constant, but could lead to a slightly lower magnitude of signal contributions in real-world examples.

It is briefly noted that similar analysis could also be conducted for vertical footprints, i.e., the sensor's penetration depth. For example, the question, at which depth groundwater rise is visible to the CRNS, could be answered by similar methods as described above, using the depth-weighting function $W_d$ instead of $W_r$ (Schrön et al., 2017). However, neutrons undergo much more interactions in the soil on their way to the detector, in strong contrast to horizontal transport, such that we suspect this endeavour to be less promising, especially for strong vertical soil moisture profiles.



### 2.4 Conversion between neutrons and soil moisture

The neutron response to each area can be calculated individually using the neutron-moisture relationship $N(\theta)$ (see Fig. 1b). However, the average (i.e., effectively measured) soil moisture, $\hat{\theta}$, may often be biased towards dryer areas due to the non-linearity of this relationship. In this study, we propose to estimate the effective soil moisture product by assuming an equally mixing neutron gas at the center of the detector, thereby following Eq. (6):

$$\hat{\theta} = \theta\left(\hat{N}\right). \tag{10}$$

The terms $N(\theta)$ and $\theta(N)$ depict the conversion functions used to derive neutrons from soil moisture and the other way round, respectively. In this study, we use the updated version of this relationship developed by Köhli et al. (2021) (Fig. 1b) which also depends on air humidity and better follows standard simulations results than the equation from Desilets et al. (2010):

$$N(\theta, h) = N_0 \left( \frac{p_1 + p_2\,\theta}{p_1 + \theta} \cdot \left(p_3 + p_4\,h + p_5\,h^2\right) \right.$$
$$\left. + e^{-p_6\,\theta}\left(p_7 + p_8\,h\right) \right), \tag{11}$$

where $N_0$ is a detector-specific scaling parameter (here: 1950 cph), $h$ is the air humidity (here: 5 g/m$^3$), and $p_{1\ldots8}$ =(0.0226, 0.207, 1.024, -0.0093, 0.000074, 1.625, 0.235, -0.0029) is the parameter set "uranos drf" from Köhli et al., which employs a CRNS-specific energy response function for typical CRNS detectors (Köhli et al., 2018). For the case of pure water, this equation reduces to: $N(\theta \to \infty) \to N_0\,p_1\,(p_3 + p_4\,h + p_5\,h^2) \approx 0.2\,N_0$.

### 2.5 Physical neutron transport simulations

Neutron transport simulations were employed, using the Monte Carlo code URANOS (Köhli et al., 2015, 2018; Köhli et al., 2021). The model setup was generated with standard layers and parameters, such as an air pressure of 1013 hPa, vertical cut-off rigidity 5 GV, domain size of $1000 \times 1000$ m, and a central cylindrical detector with 9 m radius. Neutron origins were counted as the location of the first non-air contact of a detected neutron.

Water content of the ground layer has been adapted to match the investigated soil moisture patterns. However, soil directly below and in the immediate vicinity of the detector has been kept homogeneous because the detector cannot resolve structures below its own extent. Modelled materials include soil with 50 % porosity, water (1 g/cm$^3$), concrete (2 g/cm$^3$), and in some cases an additional above-ground layer of 20 m height containing house gas (0.15 g/cm$^3$) or tree gas (0.003 g/cm$^3$). The input material definitions for all scenarios are listed in the supplement material.




## 3 Results and Discussion

### 3.1 Heterogeneous soil moisture patterns

In order to provide a reliable representation of the average soil moisture in an heterogeneous domain, it is necessary to consider the specific soil moisture conditions of each individual area. We challenge the presented concept with complex soil moisture patterns that are designed to cover difficult aspects of neutron transport for the test.

Figures 3 and 4 show soil moisture distributions at $1000\,\mathrm{m}$ and $500\,\mathrm{m}$ scales, respectively. The different areas are arranged such that fields that would theoretically contribute equally to the sensor (due to same size, distance and water content) still require their neutrons to pass other fields on their way to the detector that have much different soil moisture. The two different scales of the domain are also chosen to investigate the long-range ($r < 707\,\mathrm{m}$) and short-range ($r < 354\,\mathrm{m}$) performance of the analytical approach.

Figures 3a and 4a indicate the soil moisture pattern at the different domain scales, while the conventional footprint radius $R_{86}$ is indicated by a dashed line. Based on these hypothetical distributions of soil moisture, the individual contributions $c_i$ of each area to the neutron signal at the center (0,0) has been calculated following Eq. (7), with the results presented in in Figs. 3b and 4b. It is clearly visible that the area with $\theta = 20\,\mathrm{vol.\%}$ has the highest contribution to the signal, since it covers the direct vicinity of the detector in the center and also most of the remaining fields. As expected and in accordance with the theory, highest contribution is evident for areas that are closer to the detector and dryer than others.

We briefly showcase the calculation of the contribution of an exemplary area $A$, e.g., the bent field with $\theta_A = 50\,\mathrm{vol.\%}$ in the upper right quadrant of Fig. 4. To compute the weight of the area, one can either weight each grid cell $i$ of the matrix with $W_{r_i}/r_i$ and sum it up, or integrate $W_r$ from radii $r_1 = 98\,\mathrm{m}$ to $r_2 = 167\,\mathrm{m}$ and from angles $\vartheta_1 = 2°$ to $\vartheta_2 = 88°$. The last option is easier for radial geometries. The integration over the radii gives $0.118$ for the radial weight of a full circular ring (relative to the total weight of the domain, $\int_\Omega W_r$), while the angular weight of the circular section equals $86°/360° \approx 0.239$. This results in the normalized spatial weight of $w_A \approx 0.118 \times 0.239 = 2.7\,\%$. It would already be the sought contribution to the detector signal if the domain was homogeneous. In this heterogeneous example, however, the spatial weight needs to be multiplied with the neutrons produced by this area, $N_A = N(\theta_A = 50\,\%) \approx 651\,\mathrm{cph}$, and normalized by the effective count rate measured in the centre, $\hat{N} \approx 860\,\mathrm{cph}$, resulting in a contribution of $c_A = w_A N_A/\hat{N} \approx 2\,\%$ to the detector signal.

The results of this theoretical calculation were compared in the last step with the results of dedicated URANOS simulations. Figs. 3c and 4c show the simulated relative contribution of each area to the overall signal, where red crosses indicate the origin of neutron particles that have later hit the virtual detector. In most areas, the spatial contributions are in very good agreement with the theoretical estimations. In a few cases, the contribution of remote dry areas are underestimated which may be due to the uniform soil moisture condition of $\hat{\theta} \approx 20\,\mathrm{vol.\%}$ assumed for anchoring the radial intensity function $W_r(h, \hat{\theta})$. While the long-range transport is slightly underestimated in this case, typical scenarios are probably not as complex such that a better choice of $W_r$ could be made. Moreover, the contribution of wet fields that are arranged behind dry areas are slightly overestimated by the analytical approach, which is an effect of intermediate scattering of those neutrons on their way to the detector. While this effect is replicated in the Monte Carlo simulation, it cannot be resolved by the analytical approach.





**Figure 3.** Scenario $1000 \times 1000\,\mathrm{m}$ with a) a complex soil moisture pattern (greyscale), see also Fig. 7 for details. b) Contribution to the detector signal estimated with the analytical method, and c) simulated with URANOS.

**Figure 4.** Scenario $500 \times 500\,\mathrm{m}$ with a) a complex soil moisture pattern (greyscale), see also Fig. 7 for details. b) Contribution to the detector signal estimated with the analytical method, and c) simulated with URANOS.



Overall, the concept of estimating the signal contributions of different areas in and beyond the CRNS footprint shows a
good agreement and might be helpful for the assessment of measurement sites without rigorous neutron transport modeling.
Although higher-order corrections for interactions of the neutrons across different fields cannot be resolved with this analytical
approach, results in Figs. 3–4 indicate good overall accuracy of the estimated contributions. Where precision matters, and
under highly heterogeneous conditions (e.g., patchy snow cover), more accurate estimations may be tackled with Monte Carlo
simulations.

**3.2   Complex land-use features**

Many field sites are not only characterized by heterogeneous soil moisture patterns, but also exhibit complex land-use types,
such as tree groups, water bodies, and even urban structures (Lv et al., 2014; Iwema et al., 2015a; Schrön et al., 2018; Fersch
et al., 2020). A general view on such conditions will be provided with the following example.

This exemplary scenario consists of four regions of equal area and distance from the detector, and a fifth reference region
with the same soil moisture content as the remaining field ($\theta = 20\,\text{vol.}\%$). The five regions span over a distance from $50\,\text{m}$
to $100\,\text{m}$ and over a $45°$ arc, while they are separated by a $25°$ arc space. The land use features represented in this example
are: soil (reference area, $\theta_1 = 20\,\text{vol.}\%$), concrete pavement (equivalent to $\theta_2 = 10\,\text{vol.}\%$), a forest ($\theta_3 = 30\,\text{vol.}\%$ plus $20\,\text{m}$
tree gas), a water body ($\theta_4 \to \infty$), and a building-like structure ($\theta_5 = 10\,\text{vol.}\%$ plus $20\,\text{m}$ house gas to mimic the height of the
building).

Results shown in Fig. 5 indicate an estimated contribution of the reference area of $c_1 \approx 2.1\,\%$ (panel b), which is well
matched by the simulation, $2.3 \pm 0.3\,\%$ (panel c). The building and the concrete pavement exhibit the same dry material
composition in the ground and thus lead to similar estimated contributions, $c_2 = c_5 = 2.5\,\%$. In contrast, the simulation shows
much higher contribution of the building, $3.2 \pm 0.4\,\%$, since it accounts also for the above-ground house material. The same
holds for the forest area, $c_3 = 1.9\,\% < 2.7 \pm 0.3\,\%$. As expected, the water body shows the lowest contribution, $c_4 = 0.98\,\% <$
$0.75 \pm 0.17\,\%$.

In general, the analytical approach seems to provide good performance throughout different land use regimes, with minor
deviations at the pure-water end of the soil moisture spectrum. Significant limitations of the purely ground-driven approach are
evident for above-ground objects, such as forests or buildings. In these cases, however, the concept might still be applicable by
defining a "soil moisture equivalent" of those land use types. For example, setting $\theta_3^* = 9\,\text{vol.}\%$ and $\theta_5^* = 4\,\text{vol.}\%$ would lead
to a perfect match with the simulations for the forest and the building, respectively. While these values certainly depend on the
specific material composition and distance of the actual building or forest, future studies may show whether the contribution
of these very special land use types can be generalized.



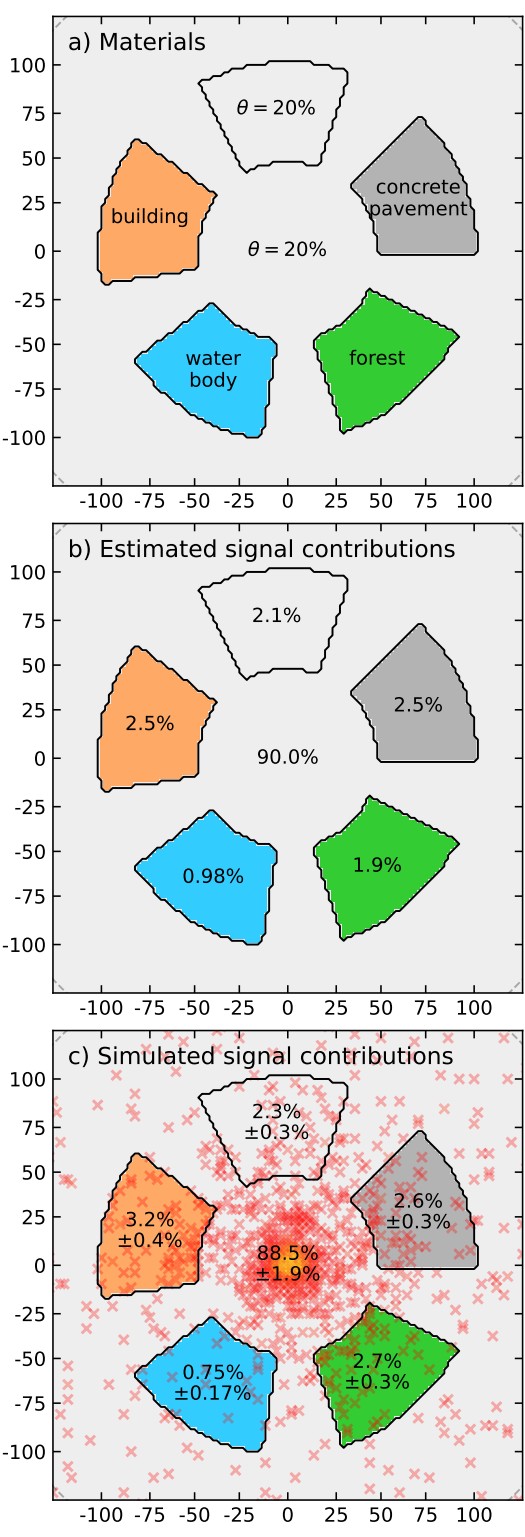

**Figure 5.** a) Exemplary scenario with different land use features (see sec. 3.2), b) analytically estimated signal contributions based only on the soil moisture, and c) URANOS simulated signal contributions including 3D features (house, trees).



### 3.3 Impact on the $N(\theta)$ relationship

The new insight that different areas in the footprint have different contribution to the finally detected signal raises the question
whether complex terrain can change the shape of the $N(\theta)$ relationship, which was initially derived from homogeneous model conditions. In fact, many authors reported deviation of their data from the standard $N(\theta)$ curve and reacted by empirically deriving site-specific parameterizations to change its shape (Rivera Villarreyes et al., 2011; Lv et al., 2014; Iwema et al., 2015b; Heidbüchel et al., 2016). In this section, we suggest that the effect of non-homogeneous signal contributions might help to explain these observations. Following the idea of the areal correction introduced by Schrön et al. (2018), we aim at
generalizing this approach to a correction based on the signal contributions.

The application of such a correction will be explained by the following theoretical example. Consider an area of interest $A_1 \in \Omega$, for which the soil moisture dynamics are to be measured by a neutron detector. A second area, $A_2 = \Omega \setminus A_1$, which does not respond to soil moisture changes, e.g., a concrete pavement, a building, a water body, a swamp, or rocky terrain lies within the sensor footprint. This area will generate a constant, invariant stream of neutrons, $N(\theta_2 = \text{const.})$, and thus dampen
the total effective neutron measurement as a function of $\theta_1$.

In order to correct for the damping effect, we propose to rescale the amplitude of the neutron counts by the signal contribution $c_1$ from area $A_1$, because only this fraction will be able to stimulate neutron dynamics:

$$\hat{N}_{\text{resc}} = \frac{N(\theta_1) - N_{\text{ref}}}{c_1} + N_{\text{ref}}, \tag{12}$$

where $N_{\text{ref}} = N(\theta_2)$ is a stationary reference offset (i.e., an invariant neutron stream from area $A_2$) around which the amplitude
will be stretched in order to make sure that the correction sustains identity for $\theta_1 = \theta_2$. If $\theta_2$ is not known, the mean observed neutron counts could be a first-order approximation, as has been done in an urban terrain by (Schrön et al., 2018).

The concept is tested in an exemplary scenario with a central area of 20 m radius and variable soil moisture, $\theta_1$, surrounded by an area of constant soil moisture of $\theta_2 = 10$ vol.% (Fig. 6a). The signal contribution of the inner area varies from 30 % to 41 % depending on $\theta_1$, with a mean of $c_1 \approx 33$ % (Eq. 7). Figure 6b shows the simulated effective neutron intensity of
the central detector as a function of $\theta_1$ (blue points), the standard $N(\theta)$ relationship (black line, see also Fig. 1b), and the function $N_{\text{resc}}$ which was rescaled by the factor $c_1$ (Eq. 12) to resemble the damping effect (blue dashed line). The shaded area represents the mentioned range of $c_1$ and demonstrates the robustness of the approach if the signal contribution cannot be determined precisely.

In summary, this application of the signal contribution theory offers an explanation for site-specific parameterizations of the
$N(\theta)$ relationship, which could be tested with existing and future CRNS data sets.



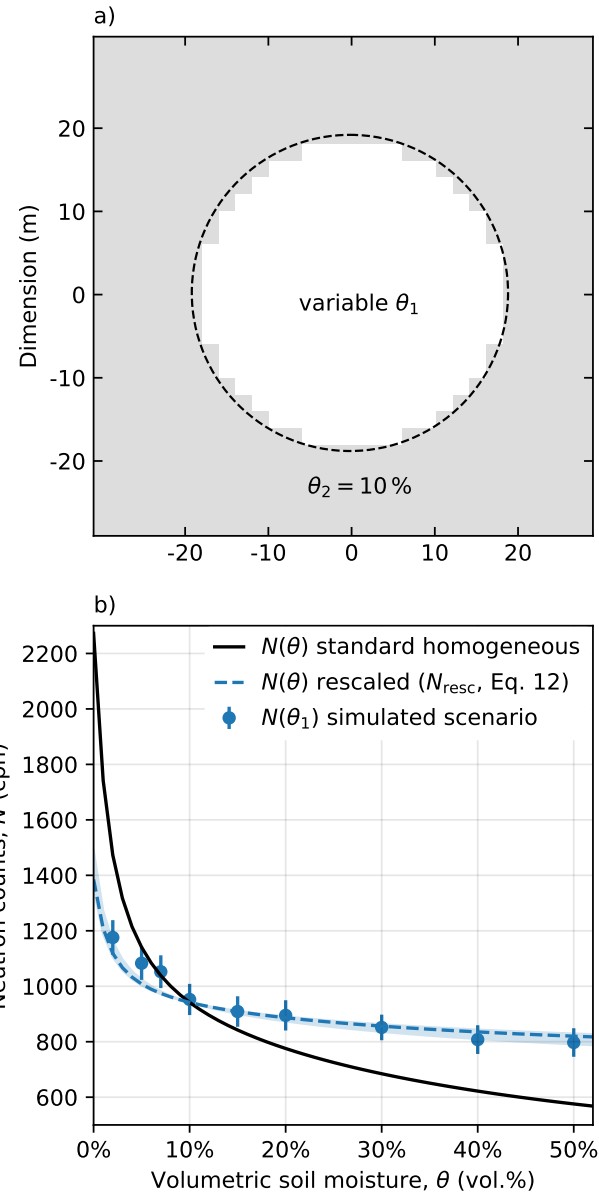

**Figure 6.** Exemplary scenario to demonstrate the impact of signal contributions on the $N(\theta)$ relationship. a) Scenario with variable soil moisture, $\theta_1$, in a central area of 20 m radius, surrounded by constant soil moisture of $\theta_2 = 10$ vol.% (e.g., concrete pavement). The signal contribution of the inner area is $c_1 \approx 33\%$. b) The simulated effective neutron intensity of the central detector as a function of $\theta_1$ (blue points) appears damped compared to the standard relationship (black line, see also Fig. 1b). However, the function can be rescaled using $c_1$ and Eq. (12) to resemble the damping effect (blue dashed line). The shaded area represents the range of $c_1$ (30–41 %) depending on soil moisture.



### 3.4 Remote field at a distance

In this section, the signal estimation concept is challenged with a more simplistic scenario, but this time without radial symmetry, in order to represent typical land use geometries. The investigated domain is split in two half spaces with different soil moisture, like two agricultural fields neighbouring each other or like partly irrigated land.

In a first exemplary scenario, the soil moisture of the two fields is set to $\theta_1 = 10$ vol.% and $\theta_2 = 30$ vol.%. According to Eqs. (6)–(7), the dry area contributes $c_1 \approx 58\%$ and the wet area $c_2 \approx 42\%$ of the total neutron count rate, while the apparent soil moisture average is $\hat{\theta} \approx 15.2$ vol.% (Eq. 10). This value is substantially lower than the naive mean, $\hat{\theta} < 20.0$ vol.% due to the non-linearity of the $\theta(N)$ relationship (Fig. 1b). URANOS simulation results confirm this approach with $\hat{\theta}_{\mathrm{sim}} \approx 14.8$ vol.%.

In order to extend the analysis to arbitrary domain splits, we now consider a scenario that consists of two areas split at the
285 distance $R$ from the center, where $A_1(x < R)$ is the field around the central detector and $A_2(x > R)$ is the remote field. The interface between the two fields is a straight line orthogonal to the $x$-axis as illustrated in Fig. 7. The total neutron count rate can be described following Eq. (6):

$$\hat{N}(\theta_1, \theta_2) = (1 - w)\, N(\theta_1) + w\, N(\theta_2), \tag{13}$$

$$\text{where} \quad w = \frac{1}{\pi} \int\limits_R^\infty W_r(h, \hat{\theta}) \arccos \frac{R}{r}\, \mathrm{d}r.$$

The weight $w$ in the Cartesian geometry is expressed in radial coordinates to avoid any corner effects, since the nature of neutron transport and detection usually follows radial symmetry. The term $\pi^{-1} \arccos R/r$ represents the length of an arc within the circle area constrained by $x > R$ (see also Fig. 8a). It can be derived from the opening angle of the sector, $\cos \alpha = R/r$, where $2\alpha$ is the same fraction of $2\pi$ as is the arc of the total circumference $2\pi r$.

In general, a purely radial geometry, where soil moisture changes in the whole region defined by $r > R$, would be a more
simple scenario to calculate. However, we consider these radial field arrangements to be a rather rarely encountered situation compared to the much more typical straight field geometries. In cases where circular fields and the corresponding soil moisture differences are relevant (e.g., for Pivot irrigation  Finkenbiner et al., 2019), the integrand can simply be solved without the $\arccos R/r$ term.

In the homogeneous case with soil moisture $\theta_1 = \theta_2$, the apparent average soil moisture also equals $\hat{\theta} = \theta_1$ and the total
300 neutron count rate results to $\hat{N}(\theta_1, \theta_2) = N(\theta_1)$. If the remote area changes from $\theta_1$ to $\theta_2$, however, the weighting function of the total domain changes slightly. The influence of this change on $W_r(h, \theta_1 \to \hat{\theta})$ is usually marginal for small changes or high distances, but the calculation could be re-iterated along updates of $\hat{\theta}$ if precision matters.

We have investigated an example scenario of a remote field at the minimal distance $R = 207$ m and soil moisture distributions of $\theta_1 = 5$ vol.% and $\theta_2 = 10$ vol.% (Fig. 7a). Equations (13) and (7) describe the influence of the remote field to the detected
neutron count rate in the center. The estimated contribution to the total neutron signal is $c_2 = w\, N(\theta_2)/\hat{N}(\theta_1, \theta_2) = 2.9\%$ (Fig. 7b) and thus significant to most CRNS detectors. Simulation results shown in Fig. 7c precisely confirm this result with 3.0%. Here, the red crosses depict the locations where detected neutrons had first contact with the soil, indicating the contribution of the corresponding region to the signal.





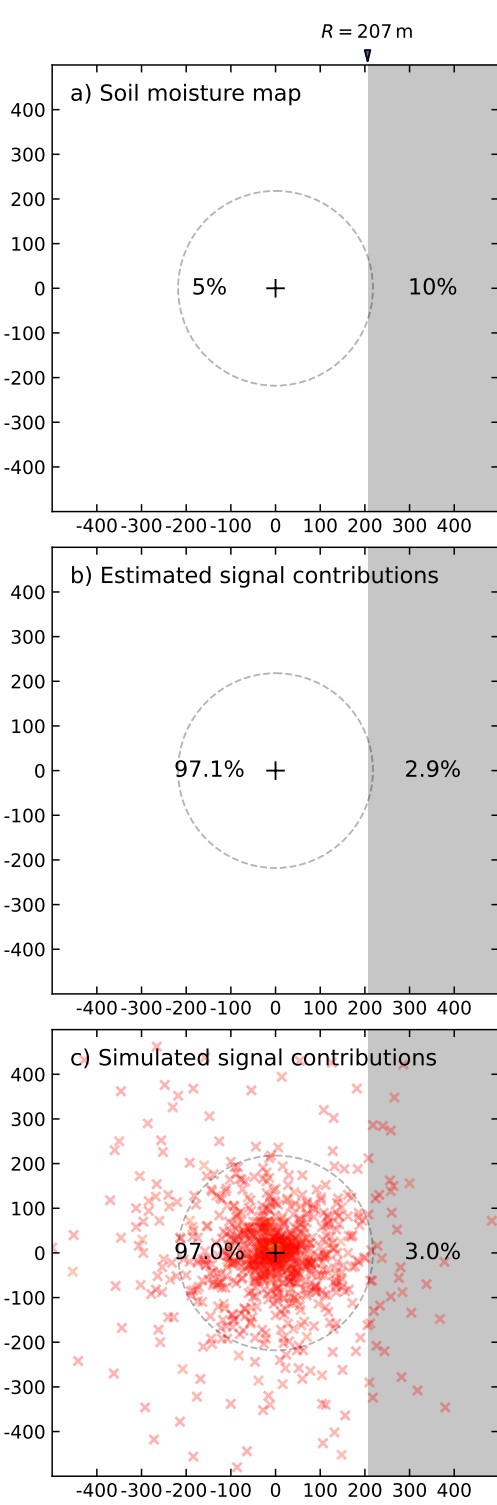

**Figure 7.** Scenario with a) soil moisture distributed in the main field ($\theta_1$, white) and in the remote field ($\theta_2$, grey) at the distance of $R = 207\,\text{m}$, the circle indicates the $R_{86}$ footprint. b) Contribution to the detector signal estimated with the analytical method, and c) simulated with URANOS (neutron origins as red crosses).



The slightly asymmetric distribution of these origins often indicate an amoeba-like shape of the footprint, suggesting that
the assumption of a symmetric footprint radius, $R_{86}$, no longer holds (as has been shown also by Köhli et al., 2015; Schattan
et al., 2019).

Interesting to note that the effectively measured soil moisture in this example is $\hat{\theta} = 5.1\,\text{vol.\%}$. Although the remote field
is very close to the outer margins of the radial footprint $R_{86}$, it still contributes $3\,\%$ to the total neutron intensity and thereby
increases the CRNS-average soil moisture by $0.1$ volumetric percent. The dry-bias can be explained by the large distance of
the wetter field (which is even larger than $R$ at all but one point), as well as by the strong non-linearity of $W_r$ towards lower $r$
(Fig. 1a), and the non-linearity of the $\theta(N)$ relationship (Fig. 1b).

In general, an exact understanding of the weighting function $W_r(h, \theta)$ plays an essential role for precise estimation of far-
field influences. This sensitivity can be illustrated using the approximation $W_r^*$ (Eq. (2)). It is usually less accurate due to
the missing dependency on air humidity and soil moisture. Considering an exemplary scenario with $\theta_1 = 5\,\text{vol.\%}$ and a field
at $R = 57\,\text{m}$ distance with $\theta_2 = 10\,\text{vol.\%}$, the "actual" signal contribution of the remote field is $c_2 = 15.3\,\%$ calculated by
URANOS. The accurate theoretical estimation using $W_r(h, \theta)$ yields $c_2 = 14.7\,\%$, while the simplified $W_r^*$ approach only
yields $11.9\,\%$.

## 3.5   A practical footprint definition based on field distance and detector sensitivity

This section proposes an alternative and more practical definition of the footprint size, by answering the following research
question: "At what distance are soil moisture changes still visible to the CRNS?", or more precisely: "At what maximum
distance $R$ from a distant field should the detector be located, such that a variation in soil moisture by $\Delta\theta = \theta_1 - \theta_2$ has
significant contribution $\Delta N \geq \sigma_N$ to the detected neutron signal?".

As has been shown in the previous sections, the intensity distribution around the sensor, $W_r(h, \theta)$, weights different regions
of the footprint in a highly inequal way. Therefore, a new approach is suggested to interpret the footprint as the distance $R$, to
which a remote change of soil moisture is still visible in the detector signal.

In order to assess this sensitivity, we reject all neutron intensity changes below a certain significance level of the sensor.
The relative stochastic precision of a neutron detector, $\sigma_N = 1/\sqrt{N}$, highly depends on the count rate $N$ (Zreda et al., 2012;
Weimar et al., 2020). It is a function of detector volume, its efficiency, atmospheric conditions, soil moisture, and temporal
aggregation (see e.g., the concept of $N_{0,\text{base}}$ in Schrön et al., 2021).

For average conditions, usual CRNS detectors with an average count rate of $N \approx 1000\,\text{cph}$ can achieve a precision of
$\sigma_N = 3.2\,\%$ per hour, or $\sigma_N/\sqrt{24} = 0.6\,\%$ per day. With regards to increasingly improving detectors and to the generally
relevant time scales of 6–12 hours, we will condition our analysis on the $\sigma_N = 1\,\%$ uncertainty limit. I.e., we will consider
CRNS detectors sensitive to a certain environmental change if the induced relative change of the count rate exceeds $1\,\%$. Note
that this definition implicates that the practical footprint $R$ may be different for different detectors, site conditions, and temporal
aggregations. Yet, as current commercial stationary systems are limited to $N \approx 5000\,\text{cph}$, this approach can be regarded as
relevant to all existing installations.



**Table 1.** Analytical results for the minimal footprint distance $R$, such that soil moisture changes of a remote field ($\theta_2 = \theta_1 + 5$ vol.%) in an initially uniform domain ($\theta_2 = \theta_1$, $h = 5\,\mathrm{g/m^3}$) become visible by the CRNS (see Fig. 7 for an illustration with $R = 207$ m). Cases consider CRNS measurement precisions of $\sigma_N = 1\%$ and $2\%$, more cases for $\sigma_N$, $h$, and $\Delta\theta$ are presented in the Supplement S1. Conventional footprints $R_{86}$ are displayed for comparison, soil moisture is displayed in vol.%. The effectively apparent soil moisture $\hat{\theta}$ is dry-biased due to the non-linearity of $\theta(N)$.

| $\theta_1$ | $\theta_2$ | $\hat{\theta}$ | $R_{(1\%)}$ | $R_{(2\%)}$ | $R_{86}$ |
|---|---|---|---|---|---|
| 1 % | 5 % | 1.1 % | 209.5 m | 147.6 m | 214 m |
| 5 % | 10 % | 5.2 % | 147.7 m | 85.4 m | 218 m |
| 10 % | 15 % | 10.4 % | 96.1 m | 40.2 m | 209 m |
| 15 % | 20 % | 15.6 % | 61.8 m | 14.4 m | 192 m |
| 20 % | 25 % | 20.7 % | 39.1 m | 3.1 m | 173 m |
| 25 % | 30 % | 25.9 % | 24.3 m | 1.2 m | 156 m |
| 30 % | 35 % | 31.0 % | 14.7 m | 0.6 m | 141 m |
| 35 % | 40 % | 36.2 % | 8.5 m | 0.2 m | 131 m |
| 40 % | 45 % | 41.3 % | 4.8 m | – | 124 m |
| 45 % | 50 % | 46.4 % | 3.0 m | – | 120 m |
| 50 % | 55 % | 51.5 % | 2.2 m | – | 115 m |

Following the above concept of the remote fields, changes of remote soil moisture conditions will only be measureable if the difference between the total neutron count rates before, $\hat{N}(\theta_1, \theta_1)$, and after the change, $\hat{N}(\theta_1, \theta_2)$, is larger than the accuracy limit:

$$\frac{\hat{N}(\theta_1, \theta_1) - \hat{N}(\theta_1, \theta_2)}{\hat{N}(\theta_1, \theta_1)} = 1 - \frac{\hat{N}(\theta_1, \theta_2)}{\hat{N}(\theta_1, \theta_1)} > \sigma_N. \tag{14}$$

Equations (13) and (14) can be solved for $R$ numerically, while an analytical solution is not straight forward due to the complexity of $W_r(h, \theta)$. To facilitate easy application of this approach for scientists and CRNS users, an interactive online tool has been developed and briefly presented in the Appendix A. For the change $\theta_1 \to \theta_2$ we suggest to use $\Delta\theta = 5$ vol.%, which is a good compromise between typical artificial or natural variations of soil moisture that are of interest for hydrologists. The supplements contain the results for $R$ using more combinations of parameters for $h$ (1–15 g/m³), $\theta_1$ (1–50 vol.%), $\Delta\theta$ (±2.5–20 vol.%), and $\sigma_N$ (1–3 %).

Calculation results of the distance $R$ are shown in Table 1 for a range of soil moisture $\theta_1$ from 1 to 50 volumetric percent, where $\theta_2$ is always larger by +5 vol.%. The measurement precision is investigated for two cases, $\sigma_N = 1\%$ and $2\%$. It is evident that the distance to the field must be much smaller if the detection precision is worse. I.e., for standard detectors at hourly resolution in humid climate it will be almost impossible to detect +5 vol.% soil moisture changes even if they occur in half of the footprint area, $R > 1$ m.





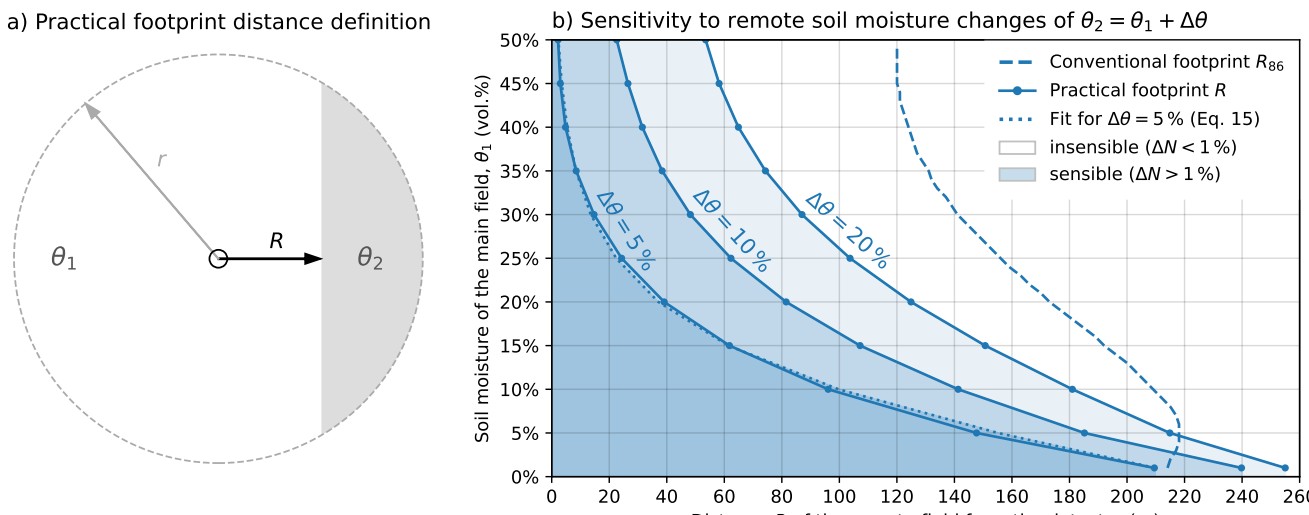

**Figure 8.** a) Schematic illustration in radial coordinates of the practical footprint definition. It represents the maximal orthogonal distance $R$ to a remote field (grey) such that its change of soil moisture from $\theta_1$ to $\theta_2 = \theta_1 + \Delta\theta$ is still sensible by the central detector. b) Main field soil moisture $\theta_1$ over distance $R$ to the remote field, for sensor precision $\sigma_N = 1\%$, air humidity $h = 5\,\text{g/m}^3$, and three wetting cases $\Delta\theta$. Assuming a main field soil moisture of 15 vol.%, for instance, an increase in remote field soil moisture by $+5$ vol.% is sensible (i.e., $\Delta N > \sigma_N$) if the distance to that field is not larger than 60 m. The dotted line approximates the practical footprint as a function of the conventional footprint radius $R_{86}$ (dashed).

Figure 8b shows the calculated ranges for $h = 5\,\text{g/m}^3$, $\sigma_N = 1\%$ and three different soil moisture changes of the remote field, $\Delta\theta = +5\ldots20$ vol.%. For arid regions between 1 and 5 percent of soil water content, the changes of $+5$ vol.% are visible to the detector at distances above 150 m. In humid climate up to 25 vol.% water content, wetter remote fields beyond 20 m distance will not show significant contribution to the detector. In wetland areas, $+5$ vol.% changes of soil moisture are hardly measureable even if this area covers almost half of the footprint. Higher soil moisture changes of $\Delta\theta = 10$ vol.% and 20 vol.% (e.g., during irrigation) are much more prominent in the neutron signal and thus allow the remote field to be at a larger distance from the sensor.

The figure also indicates the conventional footprint radius $R_{86}$ (dashed line) based on Köhli et al. (2015) and Schrön et al. (2017). Under most conditions, $R_{86}$ is much larger than $R$, as it accounts for neutron intensity changes in all directions (not only a one-sided remote field), and it has not been restricted to the mentioned accuracy limits and soil moisture changes. The radial footprint definition also fails to explain that extensive irrigation of a remote field in arid regions can be sensible much beyond the conventional footprint radius.

Similar to the wetting of the remote field, $\Delta\theta > 0$, also the case of soil drying, $\Delta\theta < 0$ has been investigated. Due to the non-linearity of $N(\theta)$, the neutron production in dry areas is disproportionally higher than the neutron reduction in wet areas. This leads to a strong influence of distant dry fields under otherwise wet conditions, which consequently manifests in longer



maximal distances $R$ by factors of $1.5$ (under wet) to $2.0$ (under dry conditions). For example, in an area with $\theta = 20\,\text{vol.\%}$, a dry-out by $-10\,\text{vol.\%}$ can be detected from a field at up to $140\,\text{m}$ distance, while a wetting of that field by $+10\,\text{vol.\%}$ is only detectable from up to $80\,\text{m}$ distance. The corresponding tabulated data is provided in the supplements.

Using a numerical fit, the new practical footprint distance $R$ can be expressed relative to the conventional footprint radius (solid lines in Fig. 8b):

$$R = R_{86}(h, \theta, P) \cdot \exp\left(0.31 - 8\theta - 5\Delta\theta\right), \tag{15}$$

where $\theta = \theta_1$ in units of $\text{m}^3/\text{m}^3$, $\Delta\theta = \pm 0.05\,\text{m}^3/\text{m}^3$, and $\sigma_N = 1\,\%$. Practical functions for higher soil moisture changes $\Delta\theta$ and higher measurement uncertainty $\sigma_N$ could be determined using the presented calculation procedure or the provided data

in the supplements.

The relative formulation based on $R_{86}$ already accounts for most dependencies on air pressure, air humidity and other factors. The equation has been tested for various air humidity conditions, for instance, and indicated good performance (not shown). If the radial footprint radius is not known, an even further simplified approximation for average air humidity $h$, standard air pressure $P$, $\Delta\theta = \pm 0.05\,\text{m}^3/\text{m}^3$, and $\sigma_N = 1\,\%$ would be:

$$R \approx 225\,\text{m} \cdot \exp\left(0.25 - 9\theta - 5\Delta\theta\right). \tag{16}$$

While these relationships may be useful to quickly assess the potential influence of distant fields on the sensor signal, we strongly encourage researchers to perform experiments (e.g., strategic irrigation) that could appropriately falsify the presented theory.

## 4   Conclusions

This paper presents an analytical concept to determine the contributions of distant areas in the footprint to the detected signal within the framework of Cosmic-Ray Neutron Sensing (CRNS). In various examples using splitted fields, heterogeneous soil moisture pattern, or complex land use types, the calculations have been verified with neutron transport simulations. The approach could be easily adapted to individual site conditions in order quantify the influence of structures, vegetated land, or irrigated fields in the footprint. The proposed method has the potential to improve sensor positioning, site-specific calibration,

and signal interpretation.

Based on this concept, two applications for the CRNS signal interpretation have been a investigated. First, found that knowledge about the signal contribution of the area of interest could help to explain seemingly site-specific shapes of the $N(\theta)$ relationship. The area's signal contribution value could be used to rescale the neutron-moisture relationship, such that the damping effect of invariant landscape features can be excluded from the signal (Fig. 6). Second, a new footprint definition has

been proposed which represents the maximum orthogonal distance to a remote field (Fig. 8a) such that its soil moisture changes are still visible in the measured neutron signal. In the presentation of the results, a typical detector precision of $\sigma_N = 1\,\%$ and positive soil moisture changes of $\Delta\theta = +5$ to $+20\,\text{vol.\%}$ have been chosen, while the approach is adaptable to any combination

of parameters. The resulting practical footprint distances for wetting remote fields are 1–90 m (wet climate), 18–180 m (humid), and 100–255 m (arid), showing the strong dependence on the initial soil moisture conditions in the field. In contrast, the

405 dry-out of remote fields ($\Delta\theta < 0$) is usually easier to detect due to the non-linearity of the neutron-water relationship, leading to 1.5–2.0× larger distances.

To date, the footprint of a CRNS sensor has been interpreted as a regular circle. The presented results show that the assumption of the radial geometry of the footprint is not suitable for very heterogeneous and complex structured regions. In fact, remote fields extending beyond the minimum tangential distance $R < x$, by this definition, usually provide less signal

contribution than radial fields beyond $R_{86} < r$. This is why $R$ is usually shorter than $R_{86}$. However, in some cases, $R$ can even be larger than $R_{86}$ for very dry regions and strong soil moisture gradients. This already indicates the low explanatory power of the radial perspective compared to the added value of the new footprint, especially for signal interpretation.

The concept could also support hydrological modeling or geostatistical inverse models, where forward-operators are required to predict the neutron intensity in a computationally efficient way. Here, the analytical calculations could facilitate spatial

neutron modeling even in complex environments without computationally expensive Monte Carlo transport simulations. In addition to first evidence provided by Schrön et al. (2018), we recommend future studies to evaluate this approach against dedicated simulations and real field data.

*Code and data availability.* Simulation data is attached as supplemental material. Analysis scripts are available as interactive Jupyter Notebooks from https://github.com/mschroen/crns-signalcontrib

*Author contributions.* MS developed the theory of signal contributions. MS, MK, and SZ developed the concept of an alternative footprint definition. MS performed the calculations and analysis. MS wrote the first version of the manuscript. MS, MK, and SZ edited and contributed to substantial improvement of the manuscript.

*Competing interests.* M. Köhli holds a CEO position at StyX Neutronica GmbH.

*Acknowledgements.* The authors thank Jannis Weimar (University of Heidelberg) for fruitful discussions. The work was funded by the DFG
(German Research Foundation) via the project 357874777, research unit FOR 2694 *Cosmic Sense* and by the German-Israeli Cooperation in Water Technology Research, BMBF project 02WIL1522.





## Appendix A: Interactive calculation of the footprint distance

To test and apply the presented concept, researchers and users may employ an easy-to-use online tool, available from: https://github.com/mschroen/crns-signalcontrib. We developed an interactive Jupyter Notebook which is hosted on GitHub and can
be run using Binder, a service that allows to run python code from the browser without installations or prior knowledge. All necessary numerical calculations related to the footprint distance and sensitivity concept are already implemented in the notebook, such that calculations of the footprint distance, signal contributions, and significance tests can be performed for user-defined soil moisture conditions (Fig. A1).

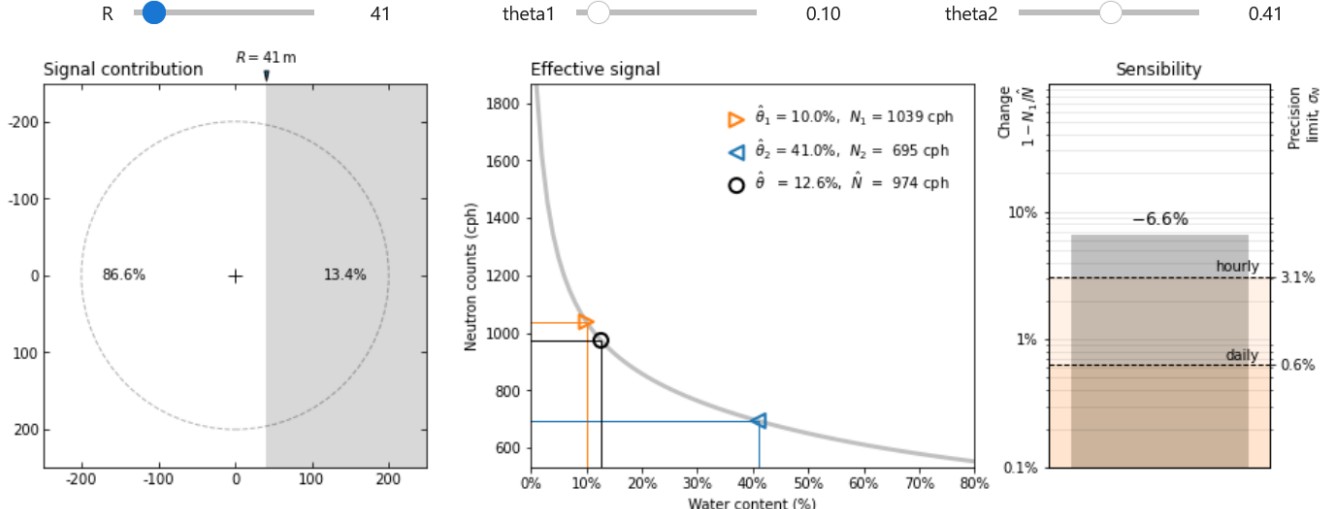

**Figure A1.** Showcase of an interactive Jupyter Notebook hosted by Binder. The tool allows to calculate the footprint distance, contributions, and significance of certain soil moisture conditions. It is accessible from the browser and does not require prior installation or programming knowledge.



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
