# Peer review of "Signal contribution of distant areas to cosmic-ray neutron sensors – implications on footprint and sensitivity"

_EGUsphere, 2022_

## Referee Comment (RC1)

Review of the manuscript Signal contribution of distant areas to cosmic-ray neutron sensors – implications on footprint and sensitivity by Schron et al.

**Summary**

The manuscript is well written and structured. The topic is relevant. The analyses well performed and discussed. The Authors have also provided an on-line tools for calculations that might be useful. I have only few main comments and some more specific/technical suggestions that are listed below. Once addressed, I think the manuscript can be suitable for publication.

**Main comments**

[1] In my opinion, one of the strongest assumption that might be warranted is that neutron contribution of each sub-domain can be assessed individually (L149). As far as I have understood, this is also actually not challenged by the MonteCarlo simulations because also in these analyses the distance of the first contact with the land is used for the calculation (L179 and L212). In contrast, assuming a more diffuse transport process I expect more interactions and mixing. This might change significantly the results. I would say that for addressing this question, a different definition of detected distance in the MonteCarlo simulation should be tested. If this will not the case within the present study, I suggest at least to extend the discussion around that.

[2] I like the practical question that has been formulated, i.e., "At what distance are soil moisture changes still visible to the CRNS" (L325). This could help better understanding neutron signal and supporting agro-hydrological applications. However, I'm not convinced that this should be posted as an alternative definition of footprint size R86 (L324). As far as I understood, on the one hand, R86 refers to the average radius over all the directions from the sensor (as stated at L365) and it does not assume that no neutrons are coming from more remote areas (on the contrary to the statement at L367). On the other hand, the Authors nicely show how soil moisture changes at a field nearby can be detected only at different distance than R86 in case soil moisture where the sensor is placed remains constant. But this does not contradict R86, i.e., on average over all direction R86 is different then the new R. Moreover, I do not have anything against these showcases but I think these support the conclusion that CRNS is not suitable for supporting irrigation management at relative small farms. But this I think was already clear from first CRNS publications. In contrast, it should be acknowledged that in any other conditions we soil experiences wetting or drying soil moisture profiles even if at different degrees that is still detected (even if in a non-linear way) by the sensor. So overall, I see the new R as an additional indicator rather than an alternative footprint size, i.e., I would still like having an indicator that accounts for neutron intensity changes in all directions (L365).

[3] my final main comment is related to the fact that all the discussion is based on a forward operator N(theta) (eq. 11). I agree that the results of this contribution will support some practical questions, e.g., where to install the sensor or modelling applications, e.g., data assimilation. However, I believe that in several applications soil moisture is the targeted variable, i.e., we do not know soil moisture within the footprint. Thus, I think one could conclude that 1) it remains an ill-posed problem to try to resolve soil moisture variability within the footprint and 2) CRNS sensor should be strategically located to avoid difficulties with signal interpretation. I would encourage the Authors to extend on these.

**Specific comments**

L2: Through the manuscript, the Authors use the term "concept" to refer to what they have developed

and tested. A concept is in my opinion an abstraction or perception. So, more than a concept, the Authors have developed an "approach" or a "method". I suggest using one of these terms instead of using "concept".

L3. I understand that similar concept (or approach, see comment before) can be developed for snow. But the study focuses on soil moisture and the snow application is not addressed. I would remove from abstract and methods the snow applications and only refer to that as a possible extension o the study in the conclusions section.

L11. "Long lasting question" is strongly CRNS community oriented. I would relax the sentence.

L34. I would not call these "problems". Maybe aspects, assumptions?

L47. More than unexpectedly, I would say that changes are expected but complicate to quantify

L55 add "soil" hydraulic conductivity to be more precise

L86. Why only humid?

L88. Vegetation height might be not a good proxy for biomass effect to the neutron signal. I guess it was selected for simplicity but would be nice to extend the discussion on the consequences of this approximation.

L141. Not sure if they rarely occur. Assuming for instance an installation at a border of the field, it may be the rule that below the sensor we have very different conditions than the surrounding areas.

L172. What is "drf"?

L193. I lost from where these ranges come from.

L249-250. As highlighted at L199, it is expected that highest contribution is evident for areas that are closer to the detector and dryer than others. What it is formulated in the present manuscript is an analytic approach to quantify these contributions. For this reason the sentence should be rephrased.

L252. The changes in the shape has also been detected for snow application. It might be worth citing and extending the discussion with that. See:

Schattan, Paul, Gabriele Baroni, Sascha E. Oswald, Johannes Schöber, Christine Fey, Christoph Kormann, Matthias Huttenlau, and Stefan Achleitner. "Continuous Monitoring of Snowpack Dynamics in Alpine Terrain by Aboveground Neutron Sensing." *Water Resources Research* 53, no. 5 (May 1, 2017): 3615–34. https://doi.org/10.1002/2016WR020234.

Schattan, P., M. Köhli, M. Schrön, G. Baroni, and S. E. Oswald. "Sensing Area-Average Snow Water Equivalent with Cosmic-Ray Neutrons: The Influence of Fractional Snow Cover." *Water Resources Research* 55, no. 12 (December 2019): 10796–812. https://doi.org/10.1029/2019WR025647.

L328. As "IT" has been shown.

---

## Referee Comment (RC4)

[referee-annotated manuscript omitted]

---

## Author Response (AR1)

**Author Response to Reviews of**

**Signal contribution of distant areas to cosmic-ray neutron sensors – implications on footprint and sensitivity**

M. Schrön, M. Köhli, S. Zacharias
*HESS*, `doi:10.5194/egusphere-2022-219`
* * *
**RC:** *Reviewer Comment*,     AR: *Author Response*,     ☐ Manuscript text

Dear Prof. Nunzio Romano,

thank you for your helpful comments and suggestions to support the second revision of our manuscript. We particularly thank you for your patience, since this revision had to be delayed due to unforeseen private circumstances.

In general, all reviewers gave positive response and supported our study mostly by further suggestions for clarifications in the text. Based on these suggestions, we substantially extended the conclusions, changed Table 1 to show $+10\%$ soil moisture changes instead of $+5\%$, and added an irregular footprint shape to Fig. 7, among other smaller changes.

For your convenience, we highlighted our key changes made to the manuscript below. All the detailed changes to the main manuscript are highlighted in a separate document.

We believe that we have adequately addressed all of the comments and hence hope that the manuscript could be accepted for publication in its current form.

On behalf of the author team,
Martin Schrön

**1. Reviewer #1**

Dear Reviewer 1,

thank you very much for your positive review, please see the full response below.

**1.1. Main comments**

**RC:** *In my opinion, one of the strongest assumption that might be warranted is that neutron contribution of each sub-domain can be assessed individually (L149). As far as I have understood, this is also actually not challenged by the MonteCarlo simulations because also in these analyses the distance of the first contact with the land is used for the calculation (L179 and L212). In contrast, assuming a more diffuse transport process I expect more interactions and mixing. This might change significantly the results. I would say that for addressing this question, a different definition of detected distance in the MonteCarlo simulation should be tested. If this will not the case within the present study, I suggest at least to extend the discussion around that.*

**AR:** *This is indeed a fundamental assumption and has been already discussed in lines 138–149 and 221–222. In fact, the question of the importance of secondary interactions beyond the individual area and the implication for the detector signal was one to which we paid particular attention in this context. That's why we challenged this hypothesis in section 3.1 with very heterogeneous soil moisture patterns. It turned out that the prediction of the contributions is still very close to Monte Carlo simulations, so the assumption, that these secondary interactions are of minor importance seems to hold. We believe this is one of the major findings of the study, so we also highlighted this in the revised conclusions.*

> The  results showed that even complex distributions of simulated neutron intensities can indeed be approximated using the new approach, indicating that secondary interactions between individual areas are of minor importance.

**RC:** *I like the practical question that has been formulated, i.e., "At what distance are soil moisture changes still visible to the CRNS" (L325). This could help better understanding neutron signal and supporting agro-hydrological applications. However, I'm not convinced that this should be posted as an alternative definition of footprint size R86 (L324). As far as I understood, on the one hand, R86 refers to the average radius over all the directions from the sensor (as stated at L365) and it does not assume that no neutrons are coming from more remote areas (on the contrary to the statement at L367). On the other hand, the Authors nicely show how soil moisture changes at a field nearby can be detected only at different distance than R86 in case soil moisture where the sensor is placed remains constant. But this does not contradict R86, i.e., on average over all direction R86 is different then the new R.*

**AR:** *There is probably a misunderstanding which we now tried to avoid in the revised version. The statement in L367 says that the definition of R86 suggests that extreme changes of soil moisture might not be "...sensible much beyond the conventional footprint radius". We are not saying that "no" neutrons are coming from beyond that radius, in fact it is implicit that 14 % of neutrons are coming from there. But the definition does not take strong soil moisture contrasts and asymmetric geometries into account. Thus, it will fail in extreme situations (as shown in Fig. 8b). Depending on the field geometry, in some directions the 86 % limit might hold, but in some it might be completely off. That is why we believe that for many applications, R has a higher practical relevance than the "on average" quantity R86.*

*We would like to point out that the original formulation of the footprint definition as R86 has already sparked discussions in the community regarding its practical applicability. The exponential shape of the radial weighting function with its very long tails and its stong near-field sensitivity is not very intuitive. Therefore providing one number for the radius led to misunderstandings regarding the actual sensitivity of a CRNS instrument. Nevertheless, R86 has its qualities, too, but it is only a radially symmetric approximation. It is certainly useful for many applications, but it is less useful in highly asymmetric cases, such as adjacent fields. On the other hand, the new R is a good alternative definition for some specific cases, but certainly it might be less useful in other situations. In the revised manuscript we made more clear that R is not a general alternative to R86, but rather more practical in certain scenarios. In general, however, the concept of defining the footprint in this, spatially more explicit way, and to include the sensor sensitivity, is applicable to any thinkable scenario.*

*Changes in section 3.5:*

> *This section proposes*  a *more practical definition of the footprint size*  for rectangular field geometries. The definition will be build upon the answer to the *following research question:*

*Changes in the conclusion:*

> *This already indicates the low explanatory power of the*  radially symmetric formulation for some situations with rectangular geometries. Here, the new cartesian footprint definition could be more informative.

**RC:** ***cnt.: Moreover, I do not have anything against these showcases but I think these support the conclusion that CRNS is not suitable for supporting irrigation management at relative small farms. But this I think was already clear from first CRNS publications. In contrast, it should be acknowledged that in any other conditions we soil experiences wetting or drying soil moisture profiles even if at different degrees that is still detected (even if in a non-linear way) by the sensor. So overall, I see the new R as an additional indicator rather than an alternative footprint size, i.e., I would still like having an indicator that accounts for neutron intensity changes in all directions (L365).***

AR: *The reviewer is correct, that for irrigated patches much smaller than the footprint the corresponding signal strength might be weak. In general, we fully agree that the new R should not replace the conventional R86 characteristics, it was not meant to. The new R should rather be an additional quantity to be more useful for some practical questions. We adapted the nuances of the text to make that more clear (see above).*

**RC:** ***my final main comment is related to the fact that all the discussion is based on a forward operator N(theta) (eq. 11). I agree that the results of this contribution will support some practical questions, e.g., where to install the sensor or modelling applications, e.g., data assimilation. However, I believe that in several applications soil moisture is the targeted variable, i.e., we do not know soil moisture within the footprint. Thus, I think one could conclude that 1) it remains an ill-posed problem to try to resolve soil moisture variability within the footprint and 2) CRNS sensor should be strategically located to avoid difficulties with signal interpretation. I would encourage the Authors to extend on these.***

AR: *We agree that we cannot directly derive soil moisture patterns from the neutron signal alone, and ideally and in the sense of a good interpretability of the signals, one should always aim at operating the sensor in areas*

*that are as homogeneous as possible. However, this requirement is often - if not in the majority of cases - not met, mainly for practical reasons. The presented method can help, firstly, to better assess the potential influence of spatial heterogeneity on the sensor signal, e.g. by forward modeling different variations of soil moisture, see how it influences the signal theoretically, and improve the uncertainty assessment of the signal. It has also the potential to support better understanding of certain features in the signal and thereby "detect" non-reported irrigation, for instance. Secondly, to drive inversion experiments (L413), i.e., to test the known variations of soil moisture patterns and find the best match to the actual soil moisture signal. This way, the subfootprint-patterns could be derived indirectly. We have clarified this in the revised manuscript:*

> In situations when sensor placement is not possible in homogeneous environments, it is crucial to realize that the sensor does not inherently provide a simple areal average of the heterogeneous soil moisture patterns in the footprint. In fact, this study showed that parts of the footprint can have different contribution to the averaged signal depending on their size and distance, while the non-linear nature of $N(\theta)$ will often underestimate the average soil moisture of two equally sized areas, as was also shown by Franz et al. (2013).
>
> To learn more about the way how a CRNS station responds to its environment, we recommend to apply the presented method by forward-modeling various soil moisture scenarios. This way, one could learn about the potential signal contributions from different landscape compartments, the implications for data uncertainty, and for hydrological process identification. The tool can also be used to quantify the challenges in signal interpretation already in the preparation phase prior to measurement campaigns. This could be an important aid with regard to the optimization of sensor placements.
>
> The  method could also support hydrological modeling or geostatistical inverse models, where forward-operators are required to predict the neutron intensity in a computationally efficient way.

**1.2. Specific comments**

**RC:** *L2: Through the manuscript, the Authors use the term "concept" to refer to what they have developed and tested. A concept is in my opinion an abstraction or perception. So, more than a concept, the Authors have developed an "approach" or a "method". I suggest using one of these terms instead of using "concept".*

**AR:** *Thank you, we have adapted the terms where necessary. In general, we believe that the "generation times transport" formula is still a theoretical concept or approach, but the way how it is applied to CRNS might indeed better be described by "method".*

**RC:** *L3. I understand that similar concept (or approach, see comment before) can be developed for snow. But the study focuses on soil moisture and the snow application is not addressed. I would remove from abstract and methods the snow applications and only refer to that as a possible extension o the study in the conclusions section.*

**AR:** *We would like to keep the few (5) occurrences of snow in the manuscript, since the implications of snow have been touched by the pure water case in section 2.4 and by the water body simulation in section 3.2. Besides irrigation, patchy snow cover is one of the most relevant fields where this method has a good potential for application. The reason is that the average descriptions of R86 and N(theta) mainly fail at the very extremes of soil moisture heterogeneity, which is one of the main take home messages of this paper. Patchy snow cover (=pure water) on rocks (=purely dry soil) hits this sweet spot, where new methods are necessary to describe the CRNS signal (see e.g., Franz et al. 2013, or Schattan et al. 2017, 2019).*

**RC:** *L11. "Long lasting question" is strongly CRNS community oriented. I would relax the sentence.*

**AR:** *We considered rephrasing the sentence, but decided to keep it as is. This is indeed a long lasting question, not only in the CRNS community, but also in remote sensing, hydrological modeling, or soil moisture monitoring, which have to address the questions of scale mismatch and representativity in comparison studies with CRNS.*

**RC:** *L34. I would not call these "problems". Maybe aspects, assumptions?*

**AR:** *Thank you, we changed to: "...the definition involves four problematic aspects".*

**RC:** *L47. More than unexpectedly, I would say that changes are expected but complicate to quantify*

**AR:** *Thank you, we changed to: "...may influence the measurement undesirably".*

**RC:** *L55 add "soil" hydraulic conductivity to be more precise*

**AR:** *Thank you, changed as suggested.*

**RC:** *L86. Why only humid?*

**AR:** *Thank you, this word was misleading and has been replaced by "homogeneous".*

**RC:** *L88. Vegetation height might be not a good proxy for biomass effect to the neutron signal. I guess it was selected for simplicity but would be nice to extend the discussion on the consequences of this approximation.*

**AR:** *Here we are simply refering to a published equation, while its vegetation height component is of almost no relevance to our study. The influence of vegetation height to the footprint has been discussed in Köhli et al. (2015). It will be interesting to better understand its influence on signal contributions in general, as well as the influence of biomass. But these topics are out of scope of this study.*

**RC:** *L141. Not sure if they rarely occur. Assuming for instance an installation at a border of the field, it may be the rule that below the sensor we have very different conditions than the surrounding areas.*

**AR:** *Thank you, we removed that statement.*

**RC:** *L172. What is "drf"?*

**AR:** *It refers to the* detector response function *as explained in the end of this sentence. We made that more clear in the revised version.*

> ...is the parameter set "uranos drf" from Köhli et al., which employs  an energy-dependent detector response function (drf) for typical CRNS  (Köhli et al. 2018).

**RC:** *L193. I lost from where these ranges come from.*

**AR:** *Sorry for the confusion. This is the Euklidian distance to the corners of a $500 \times 500$ or $200 \times 200$ square, respectively. We clarified that in the revised version.*

> The two different scales of the domain are also chosen to investigate the long-range (distance to corner: $r < 707\,\mathrm{m}$) and short-range ($r < 354\,\mathrm{m}$) performance of the analytical approach.

**RC:** *L249-250. As highlighted at L199, it is expected that highest contribution is evident for areas that are closer to the detector and dryer than others. What it is formulated in the present manuscript is an analytic approach to quantify these contributions. For this reason the sentence should be rephrased.*

AR: *Thank you for pointing this out, we removed the word "new" to clarify that this is a known and expected effect.*

**RC:** *L252. The changes in the shape has also been detected for snow application. It might be worth citing and extending the discussion with that. See: Schattan et al. 2017, Schattan et al. 2019*

AR: *Thank you for the hint. We already discussed those papers in the manuscript and have now added them also to the suggested position in the text.*

**RC:** *L328. As "IT" has been shown.*

AR: *Changed as suggested.*

**2. Reviewer #2**

**RC:** *The authors present an excellent study on the sensitivity of CRNS. They present some analytical equations to help resolve the sensitivity of CRNS footprints and transfer functions to convert neutrons into soil moisutre and vice versa. The analytical equations are validated with numerical models representing neutron transport theory. The study will be very useful for providing forward operators with low numerical costs for integrating CRNS data into hydrological or crop models for future applications. The study presents strategies for needed future work, particularly in field applications in agricultural contexts with variable irrigation. CRNS has the opportunity to be a useful technology for integration into precision agriculture. The article is well written and ready for publication following some minor corrections.*

**AR:** *Thank you for your very positive review.*

**RC:** *L184: Please explain the terms house gas and tree gas and why those are used. A bit unclear upon first read.*

**AR:** *Thank you for pointing out that we have been a bit unspecific in describing the materials used in the simulation. The terms refer to a uniform mixture of gas in the air volume which represent forests or houses, for instance. We refer to the material codes that are predefined in the URANOS model. Recently, the model code has been published on GitHub, and the corresponding detailed model description is now on GMD Discussions. In the detailed material codes description provided with the URANOS code repository, "house gas" is described as a soil-like material with 0.15 kg/m$^3$ density and 10 % water content, which roughly resemble cemented walls. "Tree gas" is defined as Cellulose (H, O, and C molecules) with 3 kg/m$^3$ density. We added these details to the revised manuscript:*

> Modelled materials include soil with 50 % porosity, water (1 g/cm³), concrete (2 g/cm³), and in some cases an additional above-ground layer of 20 m height containing  a uniform mixture of gas to represent forests or houses. The "house gas" mimics air surrounded by cement walls with soil-like material (0.15 kg/m³, 10 % water), the "tree gas" represents Cellulose with 3 kg/m³. The input material definitions for all scenarios are listed in the supplement material, see also Köhli et al. 2022 and their URANOS code repository for more details.

**3. Reviewer #3**

**RC:** *General comments: This article outlines new approaches for quantifying the effects of heterogeneities within the sensing radius of cosmic ray neutron sensors, as well as presents a new approach for estimating a non-symmetrical sensing radius. The paper is well-organized and easy to follow, and the content is novel. There are some small questions and clarifications that can be made, but after those are completed, it should be suitable for publication. Overall, I enjoyed reading this manuscript and am pleased at the advancements being made in the field of cosmic ray neutron sensing for soil moisture applications.*

**AR:** *Thank you very much for your positive review. Please see our detailed response below.*

**3.1. Specific comments**

**RC:** *The authors assume throughout the paper that the soil moisture (and the variability thereof) within a given field is known, which is not usually the case. I understand that in this instance these heterogeneities are artificially created for the purpose of testing the authors' newly developed methods, but it would be good for the authors to mention whether or not these applications are feasible without intensive soil moisture surveys.*

**AR:** *It is true that soil moisture needs to be known before the application of the concept, but this is the nature of all forward-type models, like COSMIC, MCNP, or URANOS. The great benefit from these models is that they can be used to better understand neutron data, to infer hidden hydrological processess or irrigation events, and to assess the systematic uncertainty of the data. Here, intensive soil sampling is not needed, but one could play with the soil moisture variables in the model in different parts of the footprint in order to gain a better understanding of potential influencing factors and their impact on the signal.*

*This is already indicated in the conclusion (L413), but we added a more concrete description to better communicate this in the revision. Please refer also to the reply to Reviewer 1 (RC1.3.)*

> To learn more about the way how a CRNS station responds to its environment, we recommend to apply the presented method by forward-modeling various soil moisture scenarios. This way, one could learn about the potential signal contributions from different landscape compartments, the implications for data uncertainty, and for hydrological process identification. The tool can also be used to quantify the challenges in signal interpretation already in the preparation phase prior to measurement campaigns. This could be an important aid with regard to the optimization of sensor placements.
>
> The  method could also support hydrological modeling or geostatistical inverse models, where forward-operators are required to predict the neutron intensity in a computationally efficient way.

**RC:** *Except for the last case study, the authors conduct their analyses under constant soil moisture conditions, but in reality soil moisture varies in time, which means the neutron contribution of each unique area will also change with time. This significantly complicates the cases of detected and delineated the effects of heterogeneous soil moisture patterns (section 3.1) and complex land use features (section 3.2). The authors might consider addressing this issue of temporal variability of soil moisture and whether or not the analysis carried out in the first two case studies is practical in reality.*

**AR:** *This comment is similar to the comment above, so since we made those changes shown above, we believe it is now more clear that forward-models can be used to test also temporal variability of soil moisture.*

*The generalized analytical method is aimed at being practical in reality, while the presented examples were only used to validate the analytical model with Monte Carlo physics codes. Since the validation with these specific complex scenarios was successful, we can conclude that the analytical model can be applied to any other user-defined soil moisture condition very easily. The presented framework allows users to quickly assess the neutron signal for their specific and arbitrarily varying soil moisture pattern. The attached online-notebook even provides a user-friendly interface to do exactly that. For example, the user can run the same model for low and high soil moisture conditions to gain insights on the impact of spatial contributions for dry and wet days, respectively. Hence, we hope that the presented method is as practical as possible to emulate reality.*

**RC:** *Line 56: Change "temporarily" to "temporally"*

 AR: *Changed as suggested.*

**RC:** *Line 343: Change "accuracy" to "precision"*

 AR: *Changed as suggested.*

**4. Reviewer #4**

**RC:** *This is interesting manuscript which proposes new analytical approaches for understanding cosmic-ray neutron detection in the presence of strong spatial variability in soil moisture. I commend the authors on a thoughtful new approach to this problem. The results of the proposed analytical approaches agree well with those from numerical neutron transport simulations. The results need empirical testing, as the authors have noted. The results have some practical implications for where to install (or avoid installing) cosmic-ray neutron detectors in heterogeneous environments, depending on the intended purpose of the measurements. The manuscript would be strengthed if the text focused more on explaining those practical implications and less on framing the distances calculated here as "a new practical footprint definition" for this type of detector. Because the detectors are strongly biased toward areas of dry soil, users need to understand that they do not inherently provide an accurate areal average of heterogenous footprints, as the results here show. The focus should be on choosing installation sites that minimize heterogeneity in the footprint. The results here could help inform such choices. I have included 23 specific comments, questions, and suggestions in the attached pdf version of the manuscript.*

**AR:** *Thank you very much for your positive review and the specific comments in the PDF. We have copied and responded to all comments from the annotated PDF below.*

**4.1. General comments**

**RC:** *The results have some practical implications for where to install (or avoid installing) cosmic-ray neutron detectors in heterogeneous environments, depending on the intended purpose of the measurements. The manuscript would be strengthed if the text focused more on explaining those practical implications [...] The focus should be on choosing installation sites that minimize heterogeneity in the footprint. The results here could help inform such choices.*

**AR:** *We agree that the method can help to inform the choice of installation sites, which is one of the key results. Often enough, choosing fully homogeneous installation sites is not possible, particularly in complex terrain. In those cases, the method can help to better understand the observed neutron-soil moisture relationship, to assess the systematic uncertainties produced by inhomogeneities, and to help to better identify specific events, such as remote irrigation. This is another key outcome. But understanding signal contributions is not only useful for stationary sensors: mobile CRNS is often bound to accessible roads, where sensor placement is less important than the understanding of the signal.*

*Hence, the implication on sensor placement is only one of many interesting aspects of this study. Still, we have decided to improve the discussion on these practical implications in the revised manuscript. Please see also the changes mentioned below.*

> To learn more about the way how a CRNS station responds to its environment, we recommend to apply the presented method by forward-modeling various soil moisture scenarios. This way, one could learn about the potential signal contributions from different landscape compartments, the implications for data uncertainty, and for hydrological process identification. The tool can also be used to quantify the challenges in signal interpretation already in the preparation phase prior to measurement campaigns. This could be an important aid with regard to the optimization of sensor placements.

*Please see also related changes mentioned below.*

RC: *[...] and less on framing the distances calculated here as "a new practical footprint definition" for this type of detector.*

AR: *We agree that the footprint definition should not be the main focus of the study, that is why we allocated only one of the five subsections in the 'Results' section to this topic. As discussed also with Reviewer 1, we do not intend to suggest a replacement of the conventional footprint with this new definition, but rather to suggest this new footprint concept for some specific scenarios where it could be more informative than symmetrical footprints. We clarified this more clearly in the revision in many places, e.g., also in the conclusion:*

> To date, the footprint of a CRNS sensor has been interpreted as a regular circle. The presented results show that the assumption of the radial geometry of the footprint is not suitable for very heterogeneous and complex structured regions. In fact, remote fields extending beyond the minimum tangential distance $R < x$, by this definition, usually provide less signal contribution than radial fields beyond $R_{86} < r$. This is why $R$ is usually shorter than $R_{86}$. However, in some cases, $R$ can even be larger than $R_{86}$ for very dry regions and strong soil moisture gradients. This already indicates the low explanatory power of the  radially symmetric formulation for some situations with rectangular geometries. Here, the new cartesian footprint definition could be more informative.

RC: *Because the detectors are strongly biased toward areas of dry soil, users need to understand that they do not inherently provide an accurate areal average of heterogenous footprints, as the results here show.*

AR: *Thank you for pointing this out again in a concise way. This is one of the main take-home message of our study and we used your comment, assuming your consent, as an inspiration to clarify the conclusions:*

> In situations when sensor placement is not possible in homogeneous environments, it is crucial to realize that the sensor does not inherently provide a simple areal average of the heterogeneous soil moisture patterns in the footprint. In fact, this study showed that parts of the footprint can have different contribution to the averaged signal depending on their size and distance, while the non-linear nature of $N(\theta)$ will often underestimate the average soil moisture of two equally sized areas, as was also shown by Franz et al. (2013)

**4.2. Specific comments**

RC: *Eq 2: Specify the units of "r".*

AR: *We added "(in m)".*

RC: *Eq 5: Explain how this $N(theta_i)$ is being estimated and the assumptions involved. It seems that the local neutron "production" rate for a portion of the landscape could likely be a different function than the functions often used for relating the "effective neutron intensity" to the soil moisture in the sensor's footprint (e.g. the Desilets 2010 function or similar). Please elaborate and justify any assumptions being made.*

AR: *We agree that this assumption is not trivial, but one of the outcomes of this study is that it seems to work out. The assumption that the "effective neutron intensity" of a completely homogeneous footprint behaves similar*

*to the neutron production of a portion of the landscape seems natural, though. We tried to indicate that this is just an assumption in L105 ("We propose that ..."), but we agree that this topic needs clarification.*

*We have described how $N(\theta_i)$ can be estimated in section 2.4 (Equation 11).*

*We have also discussed the involved assumption in section 2.3:*

> The  approach also treats each area individually and cannot reproduce spatial interaction effects. They can occur when neutrons generated in an area typically diffuse to nearby areas, influencing their apparent local neutron intensity (see, e.g., Schrön et al. 2018). For this reason, scenarios in this study will exhibit sufficient space between distinct areas such that their neutron contribution can be assessed individually.

*In the conclusion, we put more focus on the fact that this assumption turned out to be realistic as one of the major findings of this study:*

> In various examples using splitted fields, heterogeneous soil moisture pattern, or complex land use types, the calculations have been verified with neutron transport simulations. The  results showed that even complex distributions of simulated neutron intensities can indeed be approximated using the new approach, indicating that secondary interactions between individual areas are of minor importance.

**RC:** ***L129: "Explain what Wr and r mean exactly in the context of a grid cell. Are you making any assumptions about the size of the grid cell relative to the size of the inhomogeneous areas or to the size of the total footprint?"***

 **AR:** *Thank you for asking, $r$ is the distance to the center of a grid cell and Wr is the radial intensity at this distance. As for all numerical approximations, the size of the grid cell should be small compared to relevant structures in the footprint. Although this concept and the details already go back to Schrön et al. (2017), we agree that it deserves better explanation. We have elaborated on it more clearly in the revised text:*

> For computational implementations, it is often easier to perform calculations on Cartesian coordinate systems (the latter option), such that the individual weight of each grid cell follows $W_r/r$. Here, $r$ is the distance between the detector and the center of a grid cell and $W_r$ is the radial intensity at this distance. As for all numerical approximations, the size of the grid cell should be small compared to relevant structures in the footprint.

**RC:** *L161: Is this a known fact? If so, please provide evidence. Or is this an assumption? If so, please identify it as such.*

**AR:** *Thank you for pointing this out. It is indeed an assumption, so we changed the paragraph adequately:*

> The  measured neutron count rate $N$ of a CRNS sensor is usually estimated with a neutron-moisture relationship $N(\theta)$, where $\theta$ is the soil water content in the homogeneous sensor footprint. In this study, we postulate that this relationship can also be used to calculate the neutron intensity of each fractional area in the footprint individually (see Fig. 2b).

**RC:** *L164: footprint?*

**AR:** *Changed as suggested.*

**RC:** *Please rephrase or define exactly what is meant here.*

**AR:** *The mixing gas principle has been described in the previous section. We adapted that sentence to make that more clear:*

> Furthermore, we propose to estimate the effective soil moisture product, $\hat{\theta}$, by assuming an equally mixing neutron gas at the center of the  footprint, $\hat{N} \propto \sum N_i$, given by Eq. (6):

**RC:** *L178: "Why so large? That encompasses the majority of the sensitive zone as shown in Fig. 1."*

**AR:** *We chose 9 m radius for the detector to speed up simulations, as a reduction of the radius of the detector would drastically decrease the area of exposure to the neutrons and thus increase the simulation time. This is a standard procedure in detector simulations. If no structures were to be present below 9 m radius, the results would equally well emulate a smaller detector. For this reason, we used heterogeneous structures in our examples only beyond 9 meters. This limitation has also been discussed in L180–182.*

*It is true that this model setup would be insufficient for the scenario in Figure 1, but this is only an illustrative plot which has not been simulated in our study. Nevertheless, we added a note to the revised manuscript to clarify this:*

> The model setup was generated with standard layers and parameters, such as an air pressure of 1013 hPa, vertical cut-off rigidity 5 GV, domain size of $1000 \times 1000$ m, and a central cylindrical detector with 9 m radius. The detector size is just a numerical parameter, typically used to reduce the computational effort, and will have no impact on the results if the area below the detector is kept homogeneous.

**RC:** *L180: Unclear. Please revise and elaborate.*

**AR:** *Thank you, we changed the sentence:*

> Water content  has been added to various regions in the ground layer  in order to resemble the investigated soil moisture patterns.

**RC:** *L181: Define*

**AR:** *"direct vicinity of the detector" is a deliberate formulation in the methods section, since it is different in different scenarios investigated in the results section. In section 3.1 it is $r < 9.5\,m$, in section 3.2 it is $r < 50\,m$, and in section 3.3 it is $r < 20\,m$.*

**RC:** *L183: Please briefly define and explain here.*

**AR:** *Thank you, we added more details:*

> Modelled materials include soil with 50 % porosity, water (1 g/cm³), concrete (2 g/cm³), and in some cases an additional above-ground layer of 20 m height containing  a uniform mixture of gas to represent forests or houses. The "house gas" mimics air surrounded by cement walls with soil-like material (0.15 kg/m³, 10 g% water), the "tree gas" represents Cellulose with 3 kg/m³. The input material definitions for all scenarios are listed in the supplement material, see also Köhli et al. 2022 and their URANOS code repository for more details.

**RC:** *L249: I don't think this is a new insight. That has been known for some time. It seems that what is new here is the analytical approach to estimate those contributions. Please revise.*

**AR:** *Thank you for pointing this out, we removed the word "new" to clarify that this is a known and expected effect.*

**RC:** *L267: "I think most of the text in this section [3.3] above this point should be moved to the Methods section."*

**AR:** *The reshaping procedure is one of the results of this study and has therefore been placed in the results section. This is similar to the footprint functions in section 3.5. We believe that both these aspects are* implications *or* consequences *of the methodological concept, rather than the concept itself. So we would like to keep them outside the methods section.*

**RC:** *L306: Explain*

**AR:** *Thank you, we added the underlying calculation:*

> The estimated contribution to the total neutron signal is $c_2 = w\,N(\theta_2)/\hat{N}(\theta_1,\theta_2) = 2.9\,\%$ (Fig. 7b). This is significant to most CRNS detectors, since typical count rates of 1000 cph imply uncertainties between 0.6 % (daily) and 3.2 % (hourly resolution).

**RC:** *L309: Can you superimpose an "R86" outline based on the URANOS simulations for each of your figures with the red crosses to show this amoeba-like shape?*

**AR:** *Thank you for the suggestion, we updated Figure 7 with the suggested asymmetric footprint shape and added the corresponding text:*

> The slightly asymmetric distribution of these origins often indicate an amoeba-like shape of the footprint,  see also the red line in Fig. 7c. This suggests that the assumption of a symmetric footprint radius, $R_{86}$, no longer holds (as has been shown also byKöhli et al. 2015 and Schattan et al. 2019).

**RC:** *L312: This seems like an insignificant effect for most purposes.*

**AR:** *In this example, yes, which is exactly what we wanted to show. This is how the method can help to assess the significance of remote field irrigation, for instance.*

**RC:** *L325: minimum?*

**AR:** *It depends on the perspective. We added the word "still" to the sentence to clarify the we are looking for the maximum distance within which the remote soil water change is significant.*

> "At what maximum distance $R$ from a distant field should the detector be located, such that a  change in soil moisture by $\Delta\theta = \theta_1 - \theta_2$ still has significant contribution $\Delta N \geq \sigma_N$ to the detected neutron signal?"

**RC:** *L326: Use "difference" because "variation" often implies time.*

**AR:** *Thank you, we now use "change" instead of variation, see above.*

**RC:** *Table 1: "The chosen 5% difference in soil water content here is comparable to the magnitude of spatial variability that is commonly observed in "homogenous" fields. I think a table like this for a 10% difference in soil water content would be more instructive/educational. Differences of 10% would indicate to me an important but plausible level of inhomogeneity."*

**AR:** *Thank you for sharing your opinion about typical soil moisture changes. We agree that 10% changes would be more informative and changed the Table 1 and the corresponding text accordingly. However, we want to remind the reviewer that all thinkable combinations of parameters are also printed in a larger table in the supplements.*

> Calculation results of the distance $R$ are shown in Table 1 for a range of soil moisture $\theta_1$ from 1 to 50 volumetric percent, where $\theta_2$ is always larger by +10 vol.%. The measurement precision is investigated for two cases, $\sigma_N = 1\,\%$ and $2\,\%$. It is evident that the distance to the field must be much smaller if the detection precision is worse.  For example, standard detectors at 2-hourly resolution ($\sigma_N \approx 2\,\%$) would be able to reliably detect +10 vol.% soil moisture changes  of an adjacent field at $R = 1\,m$ distance.

[Figure]

Figure 1: (Fig. 7 in the manuscript) Scenario with a) soil moisture distributed in the main field ($\theta_1$, white) and in the remote field ($\theta_2$, grey) at the distance of $R = 207\,\text{m}$, the circle indicates the $R_{86}$ footprint. b) Contribution to the detector signal estimated with the analytical method, and c) simulated with URANOS (neutron origins as red crosses, angular footprint shape in 30° steps as red line).

Table 1: Analytical results for the minimal footprint distance $R$, such that soil moisture changes of a remote field ($\theta_2 = \theta_1 + 5$  vol.%) in an initially uniform domain ($\theta_2 = \theta_1$, $h = 5\,\text{g/m}^3$) become visible by the CRNS (see Fig. 1 for an illustration with $R = 207\,\text{m}$). Cases consider CRNS measurement precisions of $\sigma_N = 1\,\%$ and $2\,\%$, more cases for $\sigma_N$, $h$, and $\Delta\theta$ are presented in the Supplement S1. Conventional footprints $R_{86}$ are displayed for comparison, soil moisture is displayed in vol.%. The effectively apparent soil moisture $\hat{\theta}$ is dry-biased due to the non-linearity of $\theta(N)$.

| $\theta_1$ | $\theta_2$ | $\hat{\theta}$ | $R_{(1\,\%)}$ | $R_{(2\,\%)}$ | $R_{86}$ |
|---|---|---|---|---|---|
| 1 % | 11 % | 1.1 % | 239.8 m | 178.6 m | 214 m |
| 5 % | 15 % | 5.2 % | 185.2 m | 120.6 m | 218 m |
| 10 % | 20 % | 10.4 % | 141.3 m | 79.5 m | 206 m |
| 15 % | 25 % | 15.6 % | 107.2 m | 51.4 m | 189 m |
| 20 % | 30 % | 20.7 % | 81.5 m | 32.2 m | 170 m |
| 25 % | 35 % | 25.9 % | 62.3 m | 19.6 m | 150 m |
| 30 % | 40 % | 31.0 % | 48.2 m | 11.4 m | 137 m |
| 35 % | 45 % | 36.2 % | 38.4 m | 6.2 m | 127 m |
| 40 % | 50 % | 41.3 % | 31.5 m | −3.6 m | 121 m |
| 45 % | 55 % | 46.4 % | 26.5 m | −2.4 m | 120 m |
| 50 % | 60 % | 51.5 % | 22.6 m | −1.8 m | 119 m |

**RC:** *L348: Some further text and clarification is needed here. I would not use the same threshold (5%) for both a change in time (as here) and a difference in space (as previously). I agree that a change (increase/decrease over time) of soil moisture in an area by 5% is often of interest. But spatial variations of soil moisture of +/- 5% within a field are so commonplace that they are not often of special interest. For the case of spatial differences a higher threshold of something like 10% makes more sense to me.*

AR: *We agree and changed the table and the text accordingly (see above).*

**RC:** *Figure 8 caption, last sentence: This part is unclear to me. Please provide further explanation. The legend notes Eq. 15*

AR: *We have rephrased the sentence in the revised manuscript:*

> The dotted line  is an analytical formulation of $R$ (Eq. 15) as a function of the conventional footprint radius $R_{86}$ (shown as dashed line for comparison) and performs well in approximating the simulated values (points).

**RC:** *L396: This sentence is missing a subject.*

AR: *Thank you, we changed to: "First, we found ..."*

**RC:** *Appendix A: I tried running the code in Binder on Google Chrome. The code seemed to run without errors, but the figures shown below were not produced or not displayed. The sliders were visible, but not the actual figures.*

AR: *Thank you for testing the online notebook! The figures appear as soon as the sliders are used/moved. We double-checked with multiple browser and computers and confirm that it works as expected. Probably the reviewer has not attempted to move the sliders.*